# Electrothermal Modeling and Analysis of Polypyrrole-Coated Wearable E-Textiles

**DOI:** 10.3390/ma14030550

**Published:** 2021-01-24

**Authors:** Akif Kaynak, Ali Zolfagharian, Toby Featherby, Mahdi Bodaghi, M. A. Parvez Mahmud, Abbas Z Kouzani

**Affiliations:** 1School of Engineering, Deakin University, Geelong, VIC 3216, Australia; thefeathers@gmail.com (T.F.); m.a.mahmud@deakin.edu.au (M.A.P.M.); abbas.kouzani@deakin.edu.au (A.Z.K.); 2Department of Engineering, School of Science and Technology, Nottingham Trent University, Nottingham NG11 8NS, UK; mahdi.bodaghi@ntu.ac.uk

**Keywords:** conducting polymers, wearable, e-textiles, Joule heating, finite element modelling

## Abstract

The inhomogeneity of the resistance of conducting polypyrrole-coated nylon–Lycra and polyester (PET) fabrics and its effects on surface temperature were investigated through a systematic experimental and numerical work including the optimization of coating conditions to determine the lowest resistivity conductive fabrics and establish a correlation between the fabrication conditions and the efficiency and uniformity of Joule heating in conductive textiles. For this purpose, the effects of plasma pre-treatment and molar concentration analysis of the dopant anthraquinone sulfonic acid (AQSA), oxidant ferric chloride, and monomer pyrrole was carried out to establish the conditions to determine the sample with the lowest electrical resistance for generating heat and model the experiments using the finite element modeling (FEM). Both PET and nylon-Lycra underwent atmospheric plasma treatment to functionalize the fabric surface to improve the binding of the polymer and obtain coatings with reduced resistance. Both fabrics were compared in terms of average electrical resistance for both plasma treated and untreated samples. The plasma treatment induced deep black coatings with lower resistance. Then, heat-generating experiments were conducted on the polypyrrole (PPy) coated fabrics with the lowest resistance using a variable power supply to study the distribution and maximum value of the temperature. The joule heating model was developed to predict the heating of the conductive fabrics via finite element analysis. The model was based on the measured electrical resistance at different zones of the coated fabrics. It was shown that, when the fabric was backed with neoprene insulation, it would heat up quicker and more evenly. The average electrical resistance of the PPy-PET sample used was 190 Ω, and a maximum temperature reading of 43 °C was recorded. The model results exhibited good agreement with thermal camera data.

## 1. Introduction

Polypyrrole (PPy) is an organic conductive polymer which is formed by the polymerization of pyrrole by chemical or electrochemical methods [1,2]. The conductivity of PPy is influenced by chemical synthesis parameters, such as the type and molar concentration of oxidant and/or dopant, the reaction temperature and the polymerization time [3,4]. The PPy presents a broad range of temperature-dependent electrical properties, showing a resistivity decrease with the increase in temperature, similar to amorphous semiconductors [5]. The applications of intrinsically conducting polymers are limited due to the lack of processability, strength and flexibility, making them less than ideal for practical applications. However, some of these mechanical limitations can be circumvented by coating textiles with conducting polymers in order to produce fabric composites, which have the desirable mechanical properties of textiles whilst preserving the electrical properties of the conducting polymers. Durable-low resistance and highly flexible fabrics have the potential to be used in applications such as heated clothing, antistatic films, electro-magnetic shielding devices, flexible portable surface heating elements and flexible sensory equipment [6].

Amongst the intrinsically conducting polymers, PPy offers advantages over polymers, such as polyacetylene, polyindol, polythiophene and polyaniline. The chemical reaction that is done in order to oxidize pyrrole into its conductive form is much easier than the previously mentioned conductive polymers, with the pyrrole monomer being soluble and easily oxidised to produce the polymer polypyrrole. In this study, PPy was chosen due to its relatively low resistance, good electrical stability and coating fastness. Furthermore, PPy-coated fabrics also have antibacterial and antimicrobial properties, and can contribute to the fighting of antibiotic resistant pathogenic microorganisms [7]. The PPy films and PPy-coated fabrics exhibited high reflection and absorption in the microwave frequency range 1 to 10 GHz, demonstrating potential in electro-magnetic shielding applications [8,9,10,11]. The PPy-coated fabrics may also see use in medical textiles for their thermal heating properties. More recently, PPy-coated fabrics have also exhibited potential for use as sensory equipment due to electrical resistance changes with strain, aided by the flexibility of the knitted textiles. When a conducting polymer coated knitted fabric is axially stretched, there is a considerable lateral contraction which brings the yarns closer to each other thus reducing the lateral electrical resistance. This property of PPy-coated stretchable textiles can potentially be used to create flexible strain/stress sensors for a wide range of applications. Furthermore, the PPy coated fabrics have seen increasing use in equipment such as thin and lightweight fabric keyboards, pressure mats and electrocardiogram (ECG) t-shirts for the monitoring of body vitals [12]. In a recent study, a cellulosic substrate was coated with PPy through chemical polymerization followed by electrochemical deposition of copper oxide for use nitrate reduction process [13].

In two recent studies of heated fabrics, knitted cotton was used as a substrate for the in situ polymerization of pyrrole producing conductive PPy-coated cotton fabric with good breathability [2,14]. For the electrothermal heater assembly, current collectors were attached to two opposite edges of the fabric, which had shown a low initial surface resistance of 60 Ω, resulting in an average saturation temperature of approximately 50 °C at 8 V. However, in this study, Lycra and polyethyleneterephthalate (PET) are opted for rather than cotton due to their extensive use in military and outdoor applications and possessing desirable properties such as durability, wrinkle resistance, and ease of coating [12].

In a relevant earlier study, an alternative approach to the synthesis of PPy nanoparticles using hydrogen peroxide as the oxidant in a very long synthesis time of 24 h was employed [15]. The authors claim that purity of the PPy nanoparticles is ensured by using the hydrogen peroxide method in the absence of Fe ions and such nanoparticles have potential applications in drug delivery, electrochemical sensors, and fluorescence quenching-based biosensors. In the current study, however, the optimization was conducted based on the concentration of the particular set of reactants. The present study is not on nanoparticles nor the absolute purity of the PPy. This work has a materials science and engineering approach in which the optimization of coating parameters, the effect of atmospheric plasma treatment, thermal degradation, combined with a finite element modelling, based on the Joule heating model, is explored with a view to potential heating applications in wearable e-textiles.

In addition to studies on the optimization of reactant concentrations, research has been conducted on improving the surface of the substrates prior to coating [16]. One of the most effective ways of achieving this is the atmospheric plasma treatment of the substrate material prior to polymerization in order to improve the bond between the substrate and the conducting polymer coating. The substrate material is pre-treated with a plasma, consisting of a helium, argon, oxygen gas or a mixture of the three to activate the material’s surface [17]. The atmospheric pressure plasma treatment of fabrics increases the wettability of the fabric, enabling improved interactions with the coating solution [17]. The effectiveness of the adhesion of the coating to the substrates can be tested by Martindale abrasion testing, which is used to visually assess the degree of abrasion between two fabric surfaces by subjecting the coated surface to abrasion under constant weight [18].

When testing the heating properties of conductive fabrics, flexible and low resistance connections, such as very fine cabling, are preferred [19]. Contacts used in previous studies included stitching conductive wires into fabrics, for which a silicone-based wire containing fine copper strands were separated evenly and stitched on to the fabric sample to create a good electrical contact [17]. However, metallic studs used in conjunction with a cable consisting of copper strands resulted in contacts with the lowest resistance [2].

In this study, a systematic experimental and numerical work was carried out for the optimization of coating conditions to determine the lowest resistivity PPy-coated polyester and nylon fabrics and establish a correlation between the fabrication conditions and the efficiency and uniformity of Joule heating in conductive textiles. In this context, the effects of plasma pre-treatment and molar concentration analysis of the dopant anthraquinone sulfonic acid (AQSA), oxidant ferric chloride, and monomer pyrrole was systematically studied to determine the sample with the lowest resistance and uniform coating. Then the potential of the PPy coated fabric with the lowest average resistance is investigated for heating by supplying power across the fabric to produce heat and monitor the temperature by a thermal camera. The heating experiments are modelled using the finite element modeling (FEM) and by setting the relevant boundary conditions.

## 2. Methodology

### 2.1. Materials and Sample Preparation

Pyrrole (96%, Aldrich Inc., St. Louis, MO, USA) and Anthraquinone-2-sulfonic acid (AQSA) were purchased (Sigma-Aldrich, St. Louis, MO, USA) and used as received. Ferric chloride hexahydrate (FeCl_3_·6H_2_O) (98%, Fluka, Buchs, Switzerland) was used as an oxidizing agent. The samples of the nylon–Lycra and PET fabrics (Spotlight, Australia) chosen fabric were cut to 25 mm by 25 mm dimensions and coated by pyrrole through a chemical synthesis process, as shown in Figure 1. This was carried out by mixing dilute aqueous solutions of the oxidant, dopant and monomer in the reaction vessels containing the fabric samples. All reactions were carried out at room temperature (20–23 °C). Nylon–Lycra and PET fabrics were purchased from Lincraft and cut into square pieces and individually coated using a carefully selected combination of the reactant concentrations. The fabric color was chosen to be white in order to assist the initial qualitative visual assessment of the uniformity of the PPy coating and the electrical resistivity of the fabric. As the PPy coating builds up during polymerization, the white fabric gradually transitions from lighter shades of grey to mid grey and, depending on the concentrations of the reactant and the duration of polymerization, may finally result in a deep black color, which has low surface resistance. Darkening the tone of the fabric with increasing conductivity of the PPy coating enables a visual qualitative assessment of the coated fabrics prior to further evaluations.

In a previous study, Sodium Hydroxide pre-treatment was observed to have a slight improvement in the amount of PPy deposited on the fiber surface due to the increase in the surface roughness of textile fibers brought about by alkaline treatment [20]. In total, 10 g of sodium hydroxide is weighed out before being dissolved in 500 mL of deionized water; the fabric is then added to the solution, stirred and allowed to sit for 30 min while stirring occasionally. The AQSA (dopant) is firstly mixed into the 500 mL of deionized water, the iron(III) chloride hexahydrate is then crushed into a coarse powder before being added to the solution and thoroughly mixed for 5 min. The fabric is immersed into the solution prior to addition of the pyrrole monomer and mixed thoroughly. The fabric is then stirred intermittently throughout the coating time of 3 h to ensure uniformity of the coatings. Once the reaction has completed, the sample, now colored dark grey to black due to the PPy coating, is removed from the beaker, rinsed with water and left to dry.

A thin film of conducting polymer coating is deposited on the insulating fabric, which is basically a flexible fiber assembly. The non-uniform conductive coating covers the fiber surfaces and penetrates into the textile structure. Clearly, the coating on the insulating fibrous network is not in the form of a uniform flat sheet; therefore, the volume resistivity by the 4-probe method cannot accurately represent the electrical property of the coating. Therefore, the resistance measurements were carried out using the multimeter probes mounted by epoxy, separated by 1 cm with an applied fixed weight of 200 g. The contact resistance between the electrode probes and the PPy coating was minimized by using a silver conductive varnish on the coating surface (Jaycar Electronics, Sydney, Australia). Measurements were carried out systematically from the top edge at 6.25 mm intervals along the length of the sample resulting in four measurements in one row until a matrix of resistance data of 4 by 4 obtained for each sample. The same procedure was used for the resistance measurements on all the samples. Scanning electron microscopy (SEM) images of PPy coated fabric samples were performed on a Leica S440 instrument at 5 kV. The samples were gold coated prior to viewing in SEM.

### 2.2. Plasma Treatment

Atmospheric plasma produces plasma components, such as electrons, ions, free radicals and UV photons which take part in plasma-chemical reactions to initiate reactive groups and free radicals on the substrate surface, therefore, enhancing the adhesion of the coating through physical interactions in the form of secondary bonding [12].

The samples of both polyester and nylon–Lycra underwent plasma treatment to functionalize the fabric surface to improve the binding, conductivity, and stability of the PPy coating. Atmospheric plasma glow discharge (APGD) treatment on the fabrics was performed using a Sigma Technologies International APC 2000 machine. Argon gas was used at a pressure of 0.05 mbar, with a power of 56 W. The fabric samples were fed to the plasma source by a feed roller and exposed to plasma for 5 s. In total, 6 pieces of plasma-treated sample were cut, 3 each for nylon–Lycra and PET, then coated by PPy for a period of 3 h as described; washed, dried and tested for the electrical resistance of the coatings. A pair of samples were coated with the optimized parameters, another with 0.05 M and the last pair was with 0.1 M of each chemical, to note the magnitude of the effect of the plasma treatment on the electrical resistance of the coatings with different reactant concentrations.

### 2.3. Joule Heating

Joule heating defines heat dissipated by a material of nonzero electrical resistance when the electrical current is applied to it. The mathematical modeling of conductive heating fabric involves coupled physics phenomena, including electrical conduction, heat conduction with thermal power generation, and heat loss [21]. This model can describe the temperature change on the fabric surface. The general constitutive heat transfer model is used in this study, having made the assumptions that there is no coupling effect of electro physics and thermal physics, no significant thermal resistance due to contact and deformation of surfaces, the thickness of the fabric is negligible compared to the surface, and the materials parameter is constant.

For a fabric sample with the material density ρ, the heat capacity $c_{p}$, the volume v, the surface heat transfer coefficient h, the surface area A, the heater resistance R, and the applied voltage U, the heat transfer equation can be written in the form [22]:(1)$$\rho c_{p}\frac{\partial T\left(x,y,t\right)}{\partial t}-k\nabla^{2}T\left(x,y,t\right)+hAT\left(x,y,t\right)=\frac{U^{2}}{R}$$

The first two terms on the left side of (1) describe the heat flux that flows into and out of the fabric, the third term represents the heat loss due to convection from the surface, and the right side of the equation is referred to as Joule heating.

MATLAB software was used to solve the FEM. Figure 2 illustrates the boundary conditions and the equivalent discrete model of fabric conductivity. In the simulation, the free tetrahedral mesh is adopted, and the boundary conditions are defined as per the experimental conditions [23]. The material properties of the conductive heating fabric samples required for the simulation are shown in Table 1, while the ambient temperature and relative humidity are applied from experimental conditions as 20 °C and 50%. Different temperatures were applied to the left and right ends of the composite, employing a measurement setup to measure the thermoelectric properties of composites [17]. The thermal gradient was measured over a specific time using the copper electrodes. The heating bars connected to the in-house designed and fabricated temperature controllers were attached to the samples to measure the specific heat value of the fabric at room temperature and pressure. The composite properties are re-calculated based on the volume fraction of discontinuous composites [24].

Two pieces of copper tape with 1 mm size was cut and crimped on both ends of the samples being tested. Wires were then attached to the sample using an alligator clip, one attached to each copper tape. The voltage was then adjusted using the potentiometer to 5, 10, 12.5, 15 and 20 V with the temperature of the sample taken at each step using a Fluke Ti 25 thermal camera placed approximately 150 mm above the sample, and a k-type thermocouple attached to a multi-meter was used to confirm the temperature of the samples. The power supply consisted of an 18,650-lithium ion battery with a voltage of 4.2 V (fully charged), which was connected in line with a switch, a XL6009 boost convertor, allowing the voltage to be stepped up from 4.2 V to 24 V with a maximum current of 4 A. The circuit diagram is shown in Appendix A. The latter test involved the use of a 90 W wall plug power supply and the results were recorded using the thermal camera. This was done to determine the maximum achievable temperature with the equipment available.

## 3. Results and Discussions

The experiments began with some trial tests based on the information gathered from the literature and earlier works of authors on coating the fabrics by PPy [6]. Then, the optimum polymerization reactant concentrations on both nylon-Lycra and PET fabrics were determined by a systematic variation of the monomer, dopant, and oxidant concentrations and measuring the resistance of the resulting coated fabrics. Selected reactant concentrations were applied on atmospheric plasma-treated substrates and average resistance values were compared to the untreated fabrics coated using the same concentrations.

In Figure 3 and Figure 4, PPy-nylon-Lycra and PPy-PET with different coating conditions are compared. Each sample is represented by a matrix of measured resistance values using Appendix A. The sample with higher resistance is represented in a mid-grey shade, whereas the sample with lower resistance is represented in black color to emphasize the fact that the tone of the fabric deepens towards black with the increase in the conductivity of the coating.

In the first attempt, both nylon–Lycra and PET fabric samples were prepared without any chemical or plasma treatments prior to coating with 0.05 M AQSA, 0.05 M pyrrole, 0.05 M iron(III) chloride. The distribution of the resistance values in 25 mm square samples is shown in a matrix of 4 × 4 with 6.25 by 6.25 mm size elements in Figure 3. Each element represents a measured resistance data point. Based on the distribution of resistance values, it is evident that nylon–Lycra had a fairly high conductive coating, with an average resistance of 220±1.9 Ω. For the same polymerization conditions, the PPy/PET sample had a significantly lower average resistance of 82 Ω, indicating that the PPy was able to bind with the polyester more readily. This could be attributed to the differences in the surface structures of the textiles, with the PET samples being woven, more rigid and flat whereas nylon–Lycra samples knitted, flexible with tendency to drape and curl [22,24].

The outcomes of a further trial are shown in Figure 4, where both fabrics were subjected to an increased dopant concentration of 0.08 M AQSA, whilst keeping monomer and oxidant concentrations fixed at  0.05 M PPy and 0.05 M iron(III) chloride. Increasing the concentration of AQSA caused a slight decrease in the average resistance of PPy-PET and PPy-nylon–Lycra to 213±1.5 and 74±0.9 Ω, respectively. This result was expected, as it is known that increasing the amount of a dopant increases the concentration of the charge carriers in the conductive polymer, causing an increase in electrical conductivity. However, the small decrease in the magnitude of resistance is indicative of the conductivity reaching a plateau value, such that any further increase in the reactant concentrations do not cause any significant increase in the electrical conductivity but instead result in increased bulk polymerization in the solution in the form of polymer particles.

### 3.1. Molar Concentration Analysis of the Dopant AQSA, Ferric Chloride, and Pyrrole

The optimization process was continued on the PET substrates by varying one reactant at a time. The analysis was initially undertaken by fixing the concentrations of pyrrole and iron(III) chloride hexahydrate at 0.05 M and varying the AQSA from 0.01 M to a maximum value of 0.1 M. All samples were coated for a period of 3 h. All resistance results were taken at 16 points; the results were then averaged and displayed in Appendix A and Figure 5a. It was observed that resistance decreases rapidly with the increase in dopant concentration levelling off at higher dopant concentrations. Then, the molar concentrations of pyrrole and AQSA were fixed at 0.05 M and iron(III) Chloride Hexahydrate varied from 0.01 M up to 0.1 M. All samples were coated for a period of 3 h and resistance measurements were taken at 16 points, the results were then averaged and the variation of resistance with the ferric chloride concentration is displayed in Appendix A and Figure 5b. It is evident that the resistance appears to decrease with the increase in the oxidant concentration, but the reduction is much smaller in magnitude compared to the influence of the dopant concentration. Increasing the ferric chloride concentration beyond 0.1 M was observed to cause an increase in resistance. This could be attributed to overoxidation of the polypyrrole, which causes irreversible structural change. It has been observed that overoxidation is strongly influenced by the presence of nucleophiles, such as OH^−^ and Cl^−^ [25].

Finally, the analysis was undertaken by starting with a pyrrole molar concentration of 0.01 M and working up to 0.05 M in steps of 0.01 M with concentrations of both AQSA and ferric chloride fixed at 0.05 M. Again, all samples were coated for a period of 3 h and resistance readings were taken at 16 points; the results were then averaged and the change in resistance with pyrrole concentration is shown in Appendix A and Figure 5c. As in the previous variation, we see a reduction in the resistance with the increase in the monomer concentration, but the magnitude of the change is much smaller than that seen with the increase in the dopant concentration. The resistance appears to reach a plateau around 0.05 M. Increasing the pyrrole concentration beyond 0.05 M had a negligible effect on the resistance but a significant increase in bulk polymerisation was observed at higher monomer concentrations.

Upon analysis of the effect of ferric chloride concentration on the electrical resistance of coatings, it was found that increasing the concentration of the oxidant ferric chloride from 0.01 M to 0.1 M over five samples showed only a small decrease in resistance of around 25 ohms and it was concluded that a molar concentration of 0.1 M of ferric chloride would be ideal for the chemical synthesis without over oxidizing. However, increasing the molar concentration of AQSA from 0.01 M to 0.1 M had a significant impact on the electrical resistance of the PPy coating. It was seen that the coating with a concentration of 0.01 M had an average resistance of 520 Ω, which reduced to around 80 Ω at 0.1 M, a significant decrease of nearly 500 Ω. Therefore, it was decided that the AQSA concentration of 0.1 M would be adhered to for any future coatings. When the influence of the monomer concentration on resistance was examined, it was noted that increasing the molar concentration of pyrrole from 0.01 M to 0.05 M showed a slight decrease in resistance from 220 to 192 Ω. Beyond 0.05 M of pyrrole, the decrease in resistance was insignificant. Additionally, it was observed that, at higher molar concentrations, an excess of PPy was found as a black sludge in the solution due to bulk polymerization. A concentration of 0.05 M of pyrrole would be the plateau value and used for any future coatings.

When optimized concentrations were applied for the same period (3 h) it was seen that both the PET and nylon–Lycra fabric samples produced a considerably lower average resistance than previously coated samples; however, for both reactions listed in Table 2, an excess of PPy was seen as dendritic polymer particles in the solution. Polymerisation takes place both on the surface of the substrate and in the solution as conducting polymer particles, which is termed bulk polymerization. The high concentrations of the reactants, as well as long polymerization times give rise to increased bulk polymerization. Some of the bulk polymerized particles formed in the solution settle on the fabric and may also penetrate the interstices of the knitted or woven structures. These dendritic polymer particles are non-adherent to the substrate surface, they are loosely bound and readily washed off. Prolonged reaction time of 4–6 h may be considered to allow more polymerization on the fabric surface and reduce the electrical resistance of the coating. However, it should be noted that such long coating times cause excessive bulk polymerization necessitating a thorough wash, also produce coarse fabrics with thicker coatings with reduced fabric drape properties.

### 3.2. Plasma Treatment Results

Comparisons of surface morphology of the coatings in the SEM images in Figure 6 and Figure 7 for untreated and plasma-treated PPy/PET fabrics samples with identical reactant concentrations of 0.05 mol/L AQSA, 0.05 M pyrrole and 0.05 M iron(III) chloride, showed that the plasma pre-treatment increased PPy deposition and polymer growth on the surface, as it improved the wettability of the fabric (in Figure 7). This is in alignment with the observations that plasma pre-treated fabrics resulted in deep black PPy coatings, indicative of more polymer build up on the surface and reduced surface resistance [17]. Compared to the PPy coated nylon-Lycra, the PPy/PET samples had significantly lower average resistance throughout the molar concentration range for both plasma treated and untreated samples. As mentioned earlier, this could be attributed to the differences between the textile structures. The PET sample, being woven, had a more uniform flat surface with less drape, whereas the nylon–Lycra sample, being knitted with highly elastic yarns, was flexible with more drape and tendency to curl. The results in Table 3 show that the plasma treatment had the effect of decreasing the electrical resistance of all samples compared to the samples that were not plasma treated. For example, comparing plasma-treated samples #1 (Table 3) to the untreated nylon–Lycra and PET (Table 2), it was observed that the plasma-treated nylon–Lycra and PET samples had reductions in the electrical resistance by 20 and 39 Ω, respectively.

When the PET and nylon–Lycra plasma pre-treated sample #2, coated with 0.05 M of the monomer, dopant and oxidant, is compared to the plasma pre-treated samples #3, coated with 0.1 M concentrations of all the reactants, it can be seen that the resistance decreased significantly to 56 and 86 Ω, with a reduction of 26 Ω for the PET and 55 Ω for the nylon–Lycra sample (Table 3). Clearly, the lowest resistance samples are obtained when the optimum reactant concentrations are used on plasma pre-treated samples.

### 3.3. Thermal Degradation of Polypyrrole Coating

As the repeated use of conducting polymer coated fabrics for heating applications may result in premature electrical degradation due to heating, observations of the change in resistance of these PPy-coated fabrics subjected to prolonged exposure to an elevated temperature is important. Therefore, further heating experiments were conducted on a sample of plasma-treated PET fabric and a sample of untreated PET both coated at concentrations of 0.1 M AQSA, 0.05 M pyrrole, and 0.1 M iron(III) chloride hexahydrate. The initial average room temperature resistance of the PPy/PET fabrics was 62 Ω for the plasma-treated and 83 Ω for the untreated sample. The resistance of the two samples was monitored while they were subjected to accelerated thermal aging for 5 h in an oven pre-heated to 100 °C. The results are presented in Figure 8.

It was observed that, upon introducing the samples to 100 °C, the resistance initially decreased due to the rise in temperature in both samples. This is because the electrical conductivity of intrinsically conducting polymers increases with temperature, similar to amorphous semiconductors. However, after 30 min, the effects of the thermal degradation begin to be seen as a steady increase in surface resistance. Although the plasma-treated sample was slightly more stable, there was not any significant difference in the magnitude of resistance loss between the treated and untreated samples. Therefore, it should be kept in mind that exposure to elevated temperatures and/or heating the fabric by passing current through it causes electrical decay in intrinsically conducting polymer-coated fabrics.

### 3.4. Heating Conductive Fabrics

This experiment was conducted using a small variable voltage circuit running off a one-cell lithium polymer battery; a 90 W power supply was also used to test the maximum possible temperature. The average resistance of the polyester sample used was 190 Ω and the temperature readings were taken using a Fluke Ti 25 thermal imaging camera. The experiments were conducted at an ambient temperature of 22 °C and a relative humidity of 50%. The temperature results of the experiment are presented in Table 4.

Using the 90 W power supply, two thermal images were taken. Figure 9a shows the PET sample without an insulating layer on a glass support, while Figure 9b shows the same sample backed with a layer of neoprene for insulation. The effect that an insulated backing has on the materials ability to maintain its temperature can be seen in the thermal image, as well as the FEM results, in which the temperature is quite consistent across the surface of the material. An extra experiment was conducted to further demonstrate the heat loss potential of the PPy-coated PET when not backed with an insulator. A copper strip was placed underneath the heated sample to dissipate the heat from the section of the sample that it was touching. The results shown in Figure 9c would, therefore, suggest the significance of backing layers in conductive fabric design to reduce the loss of heat to surroundings and enable prolonged use with lower power inputs in practical applications. The average temperature of the heated samples in simulation and experimental results are also obtained and demonstrated in Table 5, signifying the acceptable accuracy of the FEM in this study. To further investigate the effects of fabric configuration, cross-sectional density, and insulation design on the energy efficiency, a PPy coated PET sample was heated in a rolled, tubular configuration with the 90 W power supply attached to both ends using alligator clips. Figure 10 and Table 6 reveal the effects of these design variables in achieving the maximum temperature of the conductive fabric.

The aim of the accelerated thermal degradation was to study the effect that high temperatures would have on the electrical resistance as these would have implications in real world use as heated conductive fabrics. It was found that the difference between the rate of degradation of the plasma-treated and the samples not treated by plasma was negligible and that the PPy coated textiles would not be suitable for applications where long exposure to high heat was required.

The potential of PPy-coated fabrics in heating applications was investigated by supplying power across the fabric to produce heat. A small constant current, variable voltage circuit was created in order to adjust the power supplied to the fabric from 12.5 to 50 W. The temperature across the fabric was carefully monitored by a thermal camera. The fabric’s ability to hold its heat was also tested, it was seen that, when the fabric was heated on a glass surface, the heat was quickly drained from the fabric into the glass compared to when the fabric was backed with neoprene insulation or rolled. The insulated sample would heat up quicker and more evenly than the sample without the insulation, suggesting use of layering in conductive fabrics to reduce the loss of heat to surroundings and enable prolonged use with lower power inputs in practical applications. The effect of insulated backing on the fabric’s ability to maintain its temperature was observed by thermal imaging, as well as the FEM results, which showed good agreement with the experimental data.

## 4. Conclusions

Nylon–Lycra and polyester (PET) fabric samples were coated with polypyrrole (PPy) by chemical synthesis, using an aqueous solution of pyrrole monomer, iron(III) chloride oxidant and anthraquinone sulfonic acid (AQSA) dopant in varying molar concentrations. An optimization study of the reactants determined the concentrations to be 0.05 mol/L of pyrrole, 0.1 M of iron(III) chloride hexahydrate, and 0.1 M of AQSA. Through the study of optimization of reactant concentrations of PPy coating on PET and nylon–Lycra fabrics, it was found that, regardless of the molar concentration used, the PPy would adhere to the polyester fabrics significantly better than the nylon–Lycra; this would result in the PPy/PET samples having a lower resistance even though they were coated using identical synthesis parameters.

However, the atmospheric plasma treatment improvement had the effect of decreasing the electrical resistance and improving the uniformity of the PPy coating for all reactant concentrations and substrates.

Exposure of the samples to elevated temperatures for prolonged periods caused electrical degradation. It was seen that, during an exposure of 5 h to 100 °C, both the plasma-treated and untreated samples exhibited an initial decrease in resistance, followed by a steady increase. The initial rise in conductivity is attributed to the similarity of the temperature dependence of intrinsically conducting polymers to amorphous semiconductors.

A model of joule heating was developed to predict the heating of the PPy-coated conductive fabric by finite element analysis based on the electrical resistance data at different zones of the samples. It was shown that when the fabric was backed with insulation it would heat up quicker and more evenly. FEM results exhibited good agreement with thermal camera data for each experiment.

Additionally, the effect of cross-sectional density on reducing the heat loss was noted with a sample of PPy/PET rolled into a tight tubular shape, which could be further studied. Fabric design through the backing up of the conductive surface with insulating layers, ambient temperature, input power, uniformity of the coating and initial average resistance affects the maximum temperature output of the conducting polymer-coated heated fabrics. In any case, a closed-loop control could be adopted to ensure the constant desirable temperature of wearable heated fabric over time due to the resistance change.

## Figures and Tables

**Figure 1 materials-14-00550-f001:**
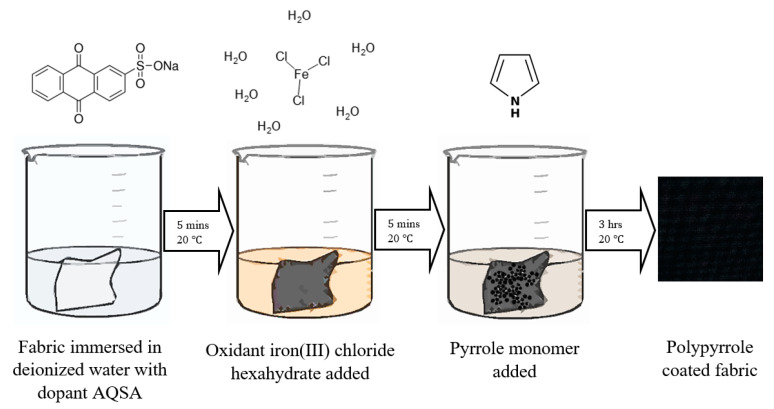
Schematic representation of preparation of the conductive fabric.

**Figure 2 materials-14-00550-f002:**
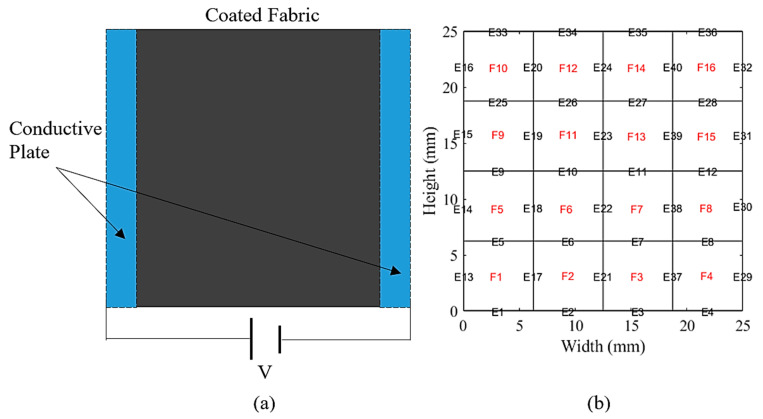
(**a**) Schematic representation of the heating fabric, and (**b**) the equivalent finite element model boundary conditions.

**Figure 3 materials-14-00550-f003:**
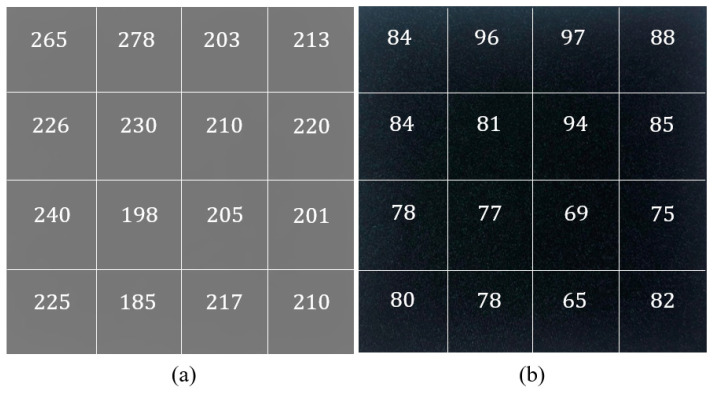
Representation of surface resistivity and the coating tone of the (**a**) nylon–Lycra and (**b**) PET fabric samples after being coated with PPy by using the reactant concentrations 0.05 M AQSA, 0.05 M pyrrole and 0.05 M iron(III) chloride.

**Figure 4 materials-14-00550-f004:**
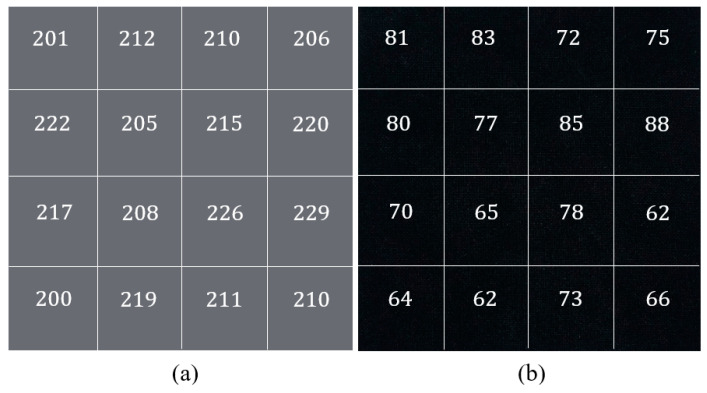
Representation of surface resistivity and the coating tone of the (**a**) nylon–Lycra, and (**b**) PET fabric samples after being coated with PPy by using the reactant concentrations 0.08 M AQSA, 0.05 M pyrrole and 0.05 M iron(III) chloride.

**Figure 5 materials-14-00550-f005:**
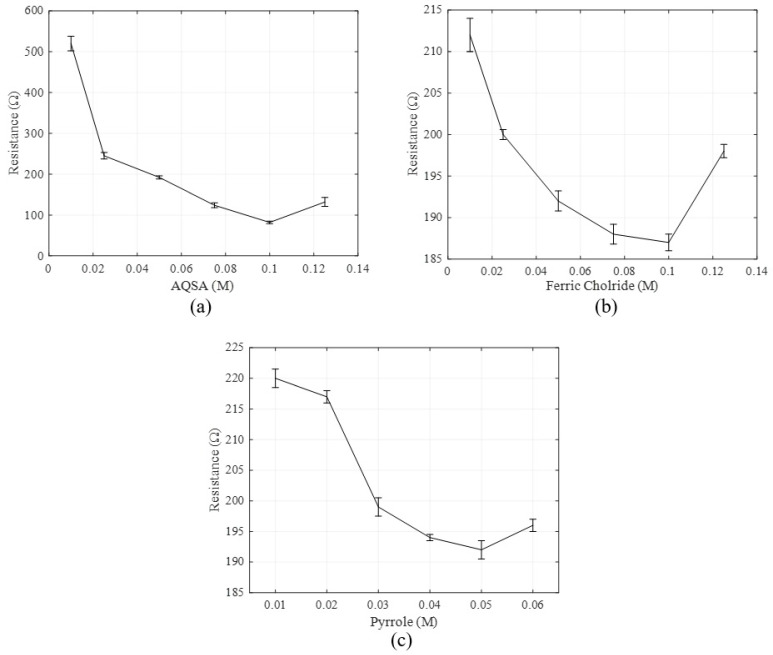
(**a**) Average resistance of fabric samples versus concentration of the dopant AQSA, (**b**) Average resistance versus the oxidant ferric/iron(III) chloride concentration, and (**c**) Average resistance versus the monomer pyrrole concentration.

**Figure 6 materials-14-00550-f006:**
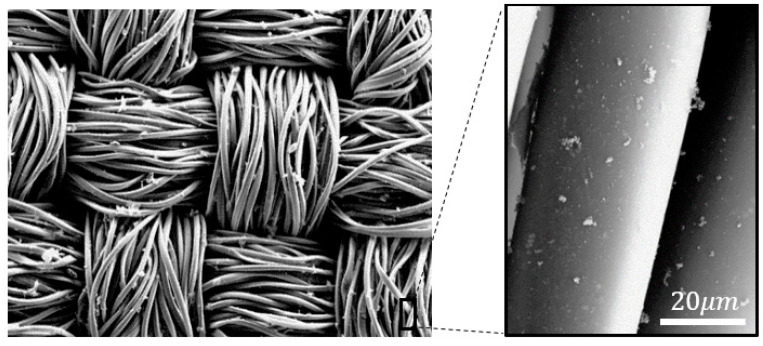
SEM images of PPy coated PET fabric without plasma treatment.

**Figure 7 materials-14-00550-f007:**
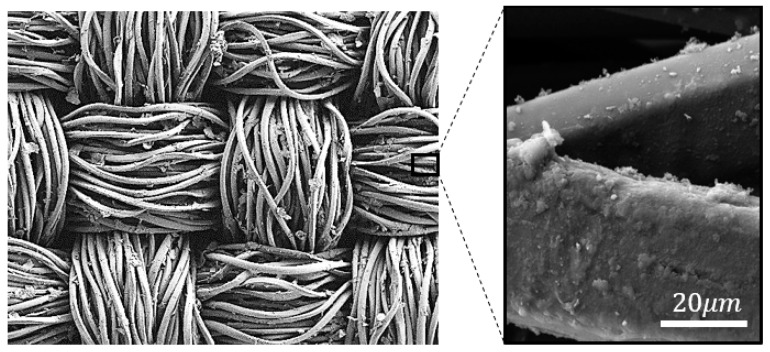
SEM image of PPy-coated PET fabric treated with plasma.

**Figure 8 materials-14-00550-f008:**
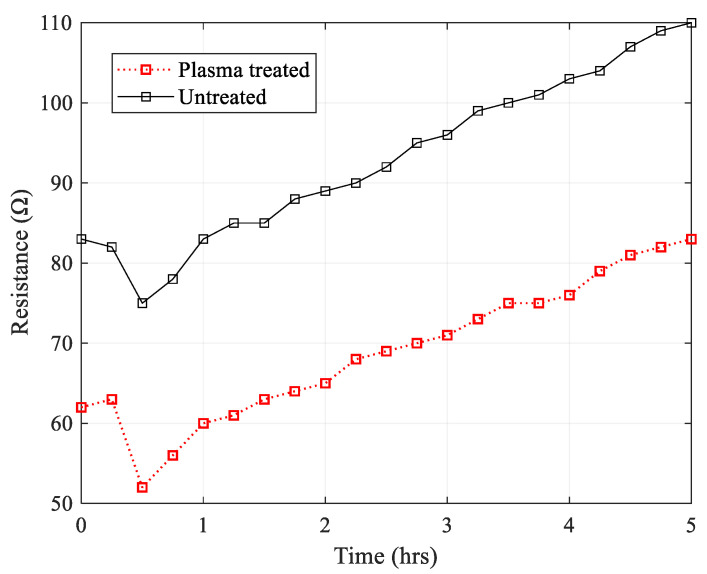
Average resistance of untreated and plasma treated PET fabric samples over 5 hrs.

**Figure 9 materials-14-00550-f009:**
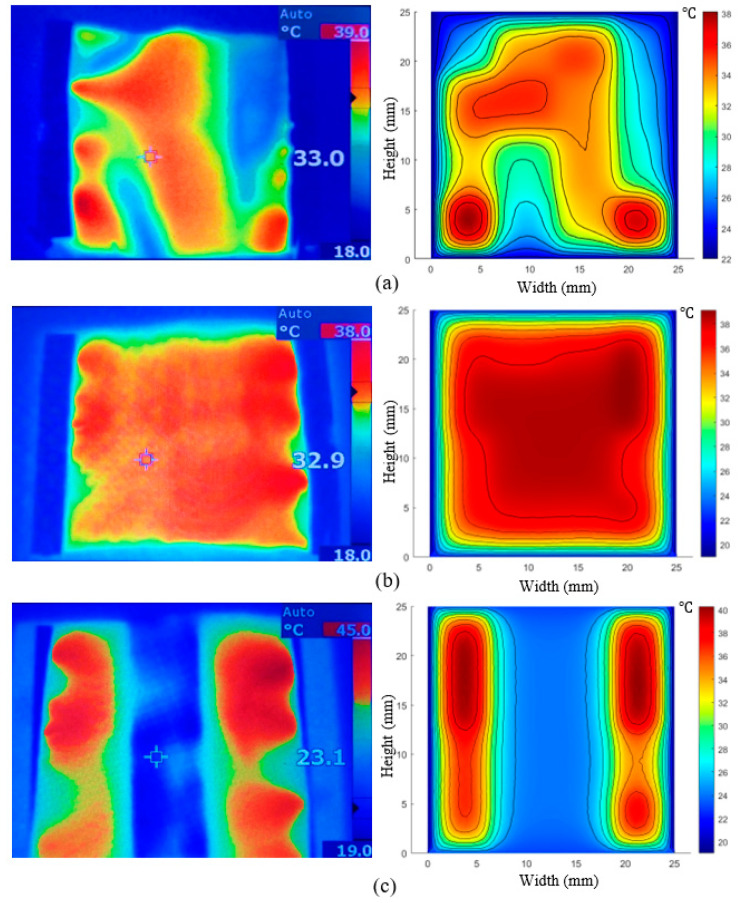
(**a**) Heated sample without insulation: thermal image (left) simulation (right), (**b**) heated sample insulated with neoprene; thermal image (left) simulation (right), (**c**) heated sample with a small strip of copper placed underneath the sample: thermal image (left) simulation (right).

**Figure 10 materials-14-00550-f010:**
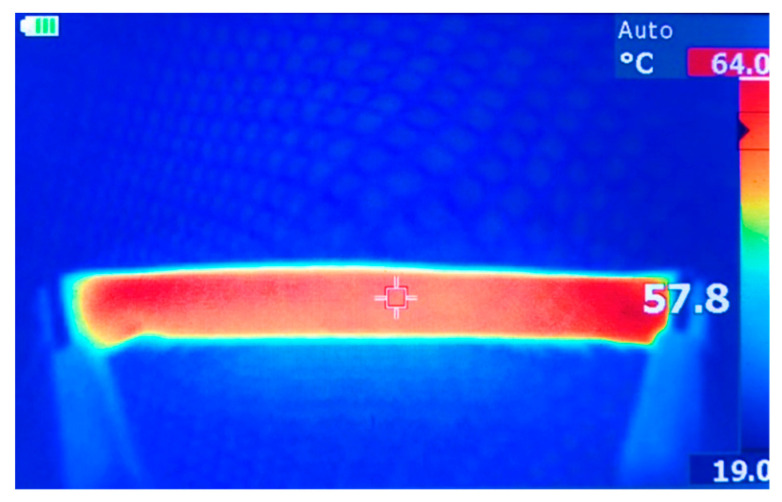
Thermal image of rolled heating test.

**Table 1 materials-14-00550-t001:** Model parameters.

Parameters	Value
**Length, mm**	25
**Width, mm**	25
**$\mathrm{Density}\;\mathrm{PPy},\;\mathrm{kg}/m^{3}$**	1150
**Thermal conductivity PPy, W/(m·K)**	50
**Thermal conductivity fabric, W/(m·K)**	0.16
**Heat capacity PPy, J/(kg·K)**	1400
**Heat capacity fabric, J/(kg·K)**	690

**Table 2 materials-14-00550-t002:** Optimized synthesis parameters results.

Sample	AQSQ (M)	Pyrrole (M)	Iron(III) Chloride Hexahydrate (M)	Resistance Average (Ω)
Polyester	0.1	0.05	0.1	82
Nylon-Lycra	0.1	0.05	0.1	141

**Table 3 materials-14-00550-t003:** Resistance values of coated fabrics with plasma pre-treatment.

Sample	AQSQ (M)	Pyrrole (M)	Iron(III) Chloride Hexahydrate (M)	Ave. Resistance (Ω)
Polyester #1	0.1	0.05	0.1	62
Nylon-Lycra #1	0.1	0.05	0.1	102
Polyester #2	0.05	0.05	0.05	75
Nylon-Lycra #2	0.05	0.05	0.05	115
Polyester #3	0.1	0.1	0.1	56
Nylon-Lycra #3	0.1	0.1	0.1	86

**Table 4 materials-14-00550-t004:** Results from heating tests for flat samples.

Test No.	Voltage (V)	Current (A)	Power (W)	Max. Temperature (°C)
1	5	2.5	12.5	25
2	10	2.5	25	28
3	12.5	2.5	31.25	32
4	15	2.5	37.5	35
5	17.5	2.5	50	38
6	20	4.5	90	43

**Table 5 materials-14-00550-t005:** Results from heating test for the rolled sample.

Sample	Ave. Temperature Exp. (°C)	Ave. Temperature Sim. (°C)	Error (%)
Without Insulation	30.07	31.12	3.50
Insulated with Neoprene	35.77	35.18	1.65
With a Copper Strip	31.08	30.45	2.03

**Table 6 materials-14-00550-t006:** Results from heating test for the rolled sample.

Test No.	Voltage (V)	Current (A)	Power (W)	Max. Temperature (°C)
1	20	2.5	50	51
2	20	4.5	90	64

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
