# Peer review of "Electrothermal Modeling and Analysis of Polypyrrole-Coated Wearable E-Textiles"

_materials, 2021, doi:10.3390/ma14030550_

Round 1
Reviewer 1 Report
The work of Akif Kaynak et all. may be published in Materials, however, after taking the following into account:
The authors covered the fabric by adding pyrroles to the layout at the end. In my opinion, this is a bad procedure, because the added pyrrole polymerizes in the entire volume of the solution and only a small part of the polymer covers the fabric fibers. In addition, polypyrrole particles can be trapped in the fabric in a "mechanical" manner, ie not taking into account the interaction of PPY-Host. The authors noticed the presence of flakes of PPY in the bulk. Depending on the type of fabric, it will be easier or more difficult to get rid of such PPY particles. This creates serious difficulties in obtaining reproducible results. For this reason, most researchers use a different procedure described, for example, in the works: Surface Review and Letters, Vol. 22, No. 5 (2015) 1550065 or POLYMER Volume 37 Number 5 1996 795.
Author Response
Reviewer 1
The work of Akif Kaynak et all. may be published in Materials, however, after taking the following into account:
The authors covered the fabric by adding pyrroles to the layout at the end. In my opinion, this is a bad procedure, because the added pyrrole polymerizes in the entire volume of the solution and only a small part of the polymer covers the fabric fibers. In addition, polypyrrole particles can be trapped in the fabric in a "mechanical" manner, ie not taking into account the interaction of PPY-Host. The authors noticed the presence of flakes of PPY in the bulk. Depending on the type of fabric, it will be easier or more difficult to get rid of such PPY particles. This creates serious difficulties in obtaining reproducible results. For this reason, most researchers use a different procedure described, for example, in the works: Surface Review and Letters, Vol. 22, No. 5 (2015) 1550065 or POLYMER Volume 37 Number 5 1996 795.
Response: The paper recommended by the reviewer entitled “Polypyrrole coated cellulosic substrate modified by copper oxide as electrode for nitrate electroreduction” which is a study of a cellulosic substrate coated with PPy through chemical polymerization followed by electrochemical deposition of copper oxide for nitrate reduction process, was included in the Introduction section and referenced.
Coating fabrics with intrinsically conducting polymers makes the fabrics conductive without any significant change in the handle or drape properties of the fabrics. Polypyrrole when doped with p-toluene sulfonic or anthraquinone sulfonic acid during the chemical polymerization gives rise to highly conductive coatings with good electrical stability. A more targeted coating method of electrochemical galvanostatic polymerization cannot be considered as the insulating fabric cannot act as an anode, therefore, cannot be coated unless the substrate already has a conductive surface. Chemical polymerization, on the other hand, can be done as either aqueous polymerization or vapor phase polymerization process. The latter gives rise to very thin coatings with high resistivity. The pre-oxidized fabric is subjected to pyrrole vapor resulting in a thin coating of polypyrrole on the fabric. Although this method circumvents deposition of bulk polymerized nodular polymer particles onto the fabric surface and the interstices of the fabric structure, the thickness of the coating is very thin and therefore the surface resistivity is high. In our paper, we employed polymerization of pyrrole in an aqueous solution with optimized concentrations of the monomer pyrrole, dopant AQSA, and oxidant ferric chloride. The method is direct, cheap, and consistent. As noted in the paper and also commented by the reviewer, the polymerization takes place both in the solution (bulk polymerization) and on the substrate surface simultaneously. The bulk polymerized PPy particles subsequently deposit onto the fabric surface and into the interstices of the textile structure (for fibrous structures) for prolonged polymerization times exceeding 2-3 hours. However, even for fabrics coated for long periods, these particles are not adherent to the fabric, they are loosely bound and easily washed off the surface without any subsequent shedding during use. We have not encountered any shedding of polymer particles after a thorough washing. Moreover, the fabrics we used were tightly woven PET with high warp and weft densities and tightly knitted Nylon-Lycra. Therefore, embedding the loose polymer particles into the textile structure was not an issue and any small fragments settled on the surface were readily washed off.
Reviewer 2 Report
In this article by Kaynak et al., the authors showed how polypyrrole coated Nylon and PET generate heat by the electrothermal effect. Then, they studied how heat distribution can be affected by certain processing measures. The theme of the article fits the scope of Materials, but there are issues, which must be tackled before this paper can be accepted for publication.
1) Please consider if citing 8 articles of the 1st author is absolutely necessary. This makes up for almost half of the references listed in this paper.
2) It is of utmost importance for a research paper to be reproducible. Otherwise, it is of limited impact as other scientists cannot build on the reported findings. Certain experimental details are missing, which makes this paper not repeatable. For instance, the size of the fabrics was not reported. More details regarding the source of these materials should also be enclosed. These are just examples of shortcomings of this sort. Please eliminate the above-mentioned and other similar errors.
3) The quality of the schemes is low and the formatting is very inconsistent. For example:
- there are three chemical compounds shown in Fig. 1 and all of them are of different size
- the beakers in the same scheme are not aligned
- the image actually contains wavy lines from MS Office under the word polypyrrole
- Fig. 2 is shrunk, font size is too low to read
- Figs. 5 and 8 are stretched and blurred
- Figs. 6 and 7 are not of the same size
- descriptors of the axes in Fig. 9 are illegible
- etc.
Please carefully go over the manuscript and improve the way the data is presented.
4) How was the issue of contact resistance eliminated? From the description of the electrical characterization methods (Lines 193 and beyond), it appears as if the samples were somehow mounted on electrical terminals made of copper without employing conductive silver paint.
5) How was the resistance measured? It seems that the authors used the 2-probe method. Please justify why the 4-probe approach was not engaged.
6) Besides that, is there a way to recalculate these values into resistivity?
7) No convincing hypothesis, which would be supported by literature or own data, was presented why a sudden increase in resistance is observed at 0.12M of ferric chloride.
8) Parameters used for SEM imaging were not included in the Experimental section
Author Response
Reviewer 2
In this article by Kaynak et al., the authors showed how polypyrrole coated Nylon and PET generate heat by the electrothermal effect. Then, they studied how heat distribution can be affected by certain processing measures. The theme of the article fits the scope of Materials, but there are issues, which must be tackled before this paper can be accepted for publication.
1) Please consider if citing 8 articles of the 1st author is absolutely necessary. This makes up for almost half of the references listed in this paper.
Response: Three of the mentioned references are removed while not affecting much the informative part of the manuscript while two of them was replaced by similar work and listed in the references as:
Hamam, A., et al., Polypyrrole coated cellulosic substrate modified by copper oxide as electrode for nitrate electroreduction. Surface Review and Letters, 2015. 22(05): p. 1550065.
Kim, M., et al., Characterization of resistive heating and thermoelectric behavior of discontinuous carbon fiber-epoxy composites. Composites Part B: Engineering, 2016. 90: p. 37-44.
2) It is of utmost importance for a research paper to be reproducible. Otherwise, it is of limited impact as other scientists cannot build on the reported findings. Certain experimental details are missing, which makes this paper not repeatable. For instance, the size of the fabrics was not reported. More details regarding the source of these materials should also be enclosed. These are just examples of shortcomings of this sort. Please eliminate the above-mentioned and other similar errors.
Response: The size of fabric is stated in the manuscript as follows:
The fabric dimensions were already mentioned at the end of section 2.1 and illustrated in Figure 2 with the resistance matrix. However, as advised by the reviewer, the source of the fabrics (Spotlight, Australia) and dimensions were included in the beginning of the ‘2. Methodology’ section and ‘2.1 sample preparation’, page 2 as:
“Pyrrole (96%, Aldrich Inc., USA) and Anthraquinone-2-sulfonic acid (AQSA) all were purchased (Sigma-Aldrich, USA) and used as received. Ferric chloride hexahydrate (FeCl3·6H2O) (98%, Fluka, Switzerland) was used as oxidizing agent. The samples of the Nylon-Lycra and PET fabrics (Spotlight Australia) chosen fabric were cut to 25 mm by 25 mm dimensions and coated by pyrrole through a chemical synthesis process as shown in Figure 1.”
Other experimental parameters were also double-checked including polymerization reactant concentrations (dopant, monomer and oxidant), temperature, humidity, polymerization time. Based on the detailed experimental parameters and directions given in the paper, the paper should be repeatable for further study.
3) The quality of the schemes is low and the formatting is very inconsistent. For example:
- there are three chemical compounds shown in Fig. 1 and all of them are of different size
- the beakers in the same scheme are not aligned
- the image actually contains wavy lines from MS Office under the word polypyrrole
- Fig. 2 is shrunk, font size is too low to read
- Figs. 5 and 8 are stretched and blurred
- Figs. 6 and 7 are not of the same size
- descriptors of the axes in Fig. 9 are illegible
- etc.
Please carefully go over the manuscript and improve the way the data is presented.
Response: Thank you for the instructions provided.
The chemical compounds in Figure 1 were adjusted to be almost the same size.
The beakers were aligned in the figure.
The word “polypyrrole’ was checked throughout the matrix for any wavy lines arising from MS office.
Figure 2 is modified and included.
Figure 5 and 8 are reconfigured in the revised manuscript.
Figure 6 and 7 are adjusted to the same size considering there are two different images.
The axes labels are added to Figure 9 simulation results.
4) How was the issue of contact resistance eliminated? From the description of the electrical characterization methods (Lines 193 and beyond), it appears as if the samples were somehow mounted on electrical terminals made of copper without employing conductive silver paint.
Response: Silver conductive varnish purchased from Jaycar was used on the fabric samples to minimize the contact resistance. The following sentence was added to the experimental section on page 4 to clarify this.
“The contact resistance between the electrode probes and the PPy coating was minimized by using a silver conductive varnish on the coating surface (Jaycar Electronics, Australia).”
5) How was the resistance measured? It seems that the authors used the 2-probe method. Please justify why the 4-probe approach was not engaged.
6) Besides that, is there a way to recalculate these values into resistivity?
Response: The use of 4-probe method for measurement is described in Supplementary Materials S1 as follows:
“In the 4-probe method a current is applied to the outer contacts of a rectangular conductive sheet of known width and thickness and the voltage drop is measured between the inner electrodes using the following
σ= (z/xy) * 1/ V (S.1)
where, s is the conductivity in S/cm
z is the distance between the inner electrodes in cm
x is the sample width in cm
y is the film thickness in cm
I is the current supplied to the outer electrodes; V is the potential difference measured between the inner electrodes.”
The inaccuracy in thickness measurement will cause significant error in the evaluation of the conductivity. When a fabric, which is a flexible fiber assembly, is coated with conducting polymer, a thin layer of PPy film is on the surface whereas the bulk of the fabric is insulating. We cannot measure the thickness of the coating accurately in this case, we do not have a rectangular conductive sheet, instead we have a conductive flexible fiber assembly with the top and bottom surfaces conductive, whereas the interior is insulating. Moreover, the fabric is compressible, and an accurate measurement of the effective thickness is not possible without causing significant error. For galvanostatically synthesized polypyrrole films 4-probe method can be used as these films are free standing sheets. Electrochemically synthesized PPy films have very limited use as they are brittle, small in size and lack the mechanical strength. Therefore for conductive fabrics the 4-probe method is not feasible, instead two probes were used. This has already been explained in the text as follows:
“A thin film of conducting polymer coating is deposited on the insulating fabric, which is basically a flexible fibre assembly. The non-uniform conductive coating covers the fiber surfaces and penetrates into the textile structure. Clearly, the coating on the insulating fibrous network is not in the form of a uniform flat sheet, therefore the volume resistivity by the 4-probe method cannot accurately represent the electrical property of the coating. Therefore, the resistance measurements were carried out by using the multimeter probes mounted by epoxy, separated by 1 cm with an applied fixed weight of 200 g. The contact resistance between the electrode probes and the PPy coating was minimized by using a silver conductive varnish on the coating surface (Jaycar Electronics, Australia). Measurements were carried out systematically from the top edge at 6.25 mm intervals along the length of the sample resulting in four measurements in one row until a matrix of resistance data of 4 by 4 obtained for each sample. The same procedure was used for the resistance measurements on all the samples.”
7) No convincing hypothesis, which would be supported by literature or own data, was presented why a sudden increase in resistance is observed at 0.12M of ferric chloride.
Response: The following is included in the revised manuscript:
“Increasing the ferric chloride concentration beyond 0.1 M was observed to cause increase in resistance. This could be attributed to overoxidation of the polypyrrole, which causes irreversible structural change. It has been observed that overoxidation is strongly influenced by the presence of nucleophiles such as OH- and Cl- [24].”
8) Parameters used for SEM imaging were not included in the Experimental section
Response:
The following paragraph was added to the experimental section:
“Scanning electron microscopy (SEM) images of polypyrrole coated fabric samples were performed on a Leica S440 instrument at 5kV. The samples were gold coated prior to viewing in SEM.”
Reviewer 3 Report
The paper exhibited detailed analysis of optimization the coating of polypyrrole (PPy) on polymer fabrics to enhance the materials conductivity and investigated the effects of the coating on the surface temperature. It could be a reference for massive production of conductive fabrics. It is recommended for publication after consider the comments:
Line 123. please add citation why using NaOH.
Line 124. Please use SI units and symbols throughout the manuscript, for example, “gr” should be “g”
Line 163. Figure 1. For each arrow, you can add operation conditions. Temperature (though ambient), time of the treatment. Etc.
Line 177 Equation 1. Please delete the repeated “(1)”
Line 180 Any sources/citations or considerations that you use to determine these boundary conditions?
Line 207 Figure 2. Is there a physical meaning that the plate is divided into 16 segments or just randomly chosen?
Line 287 to Line 290. Any physical characterization or testing results that can support this claim?
Line 385 to Line 399. It would be better to put these sentences into related sections, for example, “It was found … without over oxidizing” would be after Line 354.
General comments: Is the found optimal concentrations of the chemicals, that used to coat PPy on the two polymer Nylon and PET Fabrics, universal?
Author Response
Reviewer 3
The paper exhibited detailed analysis of optimization the coating of polypyrrole (PPy) on polymer fabrics to enhance the materials conductivity and investigated the effects of the coating on the surface temperature. It could be a reference for massive production of conductive fabrics. It is recommended for publication after consider the comments:
Line 123. please add citation why using NaOH.
Response: The following section is included in the manuscript:
Sodium Hydroxide pre-treatment appears to have a slight improvement in the amount of polypyrrole deposited on the fiber surface increased due to the increase in the surface roughness of textile fibers brought about by alkaline treatment. In a previous study, AFM images showed a smooth surface morphology on the untreated fiber whereas the treated fiber had a higher surface roughness, improving the deposition of the polypyrrole to the fiber surface, which in turn had a positive influence in the electrical properties. The following is included in the manuscript:
In a previous study Sodium Hydroxide pre-treatment was observed to have a slight improvement in the amount of polypyrrole deposited on the fiber surface due to the increase in the surface roughness of textile fibers brought about by alkaline treatment [19].
Line 124. Please use SI units and symbols throughout the manuscript, for example, “gr” should be “g”
Response: The SI unit symbols are checked throughout the manuscript, including “g”.
Line 163. Figure 1. For each arrow, you can add operation conditions. Temperature (though ambient), time of the treatment. Etc.
Response: Thank you for the suggestion. The mentioned information is included in Figure 1.
Line 177 Equation 1. Please delete the repeated “(1)”
Response: Thank you for the notice, the repeated (1) is removed.
Line 180 Any sources/citations or considerations that you use to determine these boundary conditions?
Response: The MATLAB software was used to solve the FEM and reference [18] was referred for developing the boundary conditions of Joule heating. These are included in the manuscript.
Line 207 Figure 2. Is there a physical meaning that the plate is divided into 16 segments or just randomly chosen?
Response: The number of the segments are the function of the size of fabric and feasibility of experiment to measure the conductivity of fabric surface. The choice was made considering evenly spreading out the resistance measurements through the fabric surface, while taking into account the fixed distance between the electrodes of the multimeter.
Line 287 to Line 290. Any physical characterization or testing results that can support this claim?
Response: Having applied the optimized concentrations, it was seen that both samples produced a considerably lower average resistance than previously tested samples, however for both reactions listed in Table 2, it was noted that an excess of PPy was seen as dendritic polymer particles in the solution. Polymerisation takes place both on the surface of the substrate and in the solution, which is termed as bulk polymerization. The high concentrations of the reactants as well as long polymerization times give rise to excessive bulk polymerization.
It is known that in the chemical polymerization of pyrrole the polymerization takes place both in the solution (bulk polymerization) and on the substrate surface simultaneously. The bulk polymerized PPy particles in solution subsequently settle down and deposit onto the fabric surface and may also penetrate the interstices of the textile structure (for fibrous textile structures) for prolonged polymerization times exceeding 2-3 hours. However, even for fabrics coated long periods, these particles are not adherent to the fabric, they are loosely bound and easily washed off the surface without any subsequent shedding during use. We have not encountered any shedding of polymer particles after a thorough washing. Moreover, the fabrics we used were tightly woven PET with high warp and weft densities and tightly knitted Nylon-Lycra. Therefore, embedding of the loose polymer particles into the textile structure was not an issue and any small fragments settled on the surface was readily washed off. Increased bulk polymerization could be visually observed in the solution and dendritic polymer particles settled on the surface of the fabric which were non-adherent to the surface. This was reported in detail with supporting SEM images in an earlier study “Kaynak, Akif and Beltran, Rafael 2003, Effect of synthesis parameters on the electrical conductivity of polypyrrole-coated poly(ethylene terephthalate) fabrics, Polymer international, vol. 52, no. 6, pp. 1021-1026.”
The reviewer may be referring to the following lines:
“Comparisons of surface morphology of the coatings in the SEM images in Figures 6 and 7 for untreated and plasma-treated PPy/PET fabrics samples with identical reactant concentrations of 0.05 mol/L AQSA, 0.05 M pyrrole and 0.05 M iron(III) chloride, showed that the plasma pre-treatment improved the wettability of the fabric and led to increased polypyrrole deposition and polymer growth on the surface (in Figure 7). This is in alignment with the observations that plasma pre-treated fabrics resulted in deep black PPy coatings, indicative of more polymer build up on the surface and reduced surface resistance [16].”
Please note that references added to support the observations in this section.
Line 385 to Line 399. It would be better to put these sentences into related sections, for example, “It was found … without over oxidizing” would be after Line 354.
Response: This was an important observation, that improved the flow of the article. The paragraph mentioned was moved to section 3.1. After moving the paragraph, the manuscript was read again and further editing and positioning of the graphs and some paragraphs made to improve the readability and flow of the paper.
General comments: Is the found optimal concentrations of the chemicals, that used to coat PPy on the two polymer Nylon and PET Fabrics, universal?
Response: The optimization of the concentration of the reactants (pyrrole monomer, oxidant ferric chloride and the dopant anthraquinone sulfonic acid (AQSA) were carried out at controlled temperature and humidity in exactly the same manner for each coating. The reactant concentrations were systematically increased, one at a time, while keeping the others constant. In this paper all the experimental details are revealed and by following the instructions given in the manuscript the coatings should be repeatable and therefore results are universal. For future trials that may include different experimental conditions and reactants, a systematic optimization study should be conducted to determine the desired quality of coatings with respect to resistance, surface morphology, fabric color and drape properties.
Round 2
Reviewer 1 Report
Last version can be published.
Author Response
Thank you!
Reviewer 2 Report
Thank you.
I recommend the publication of the article in the present form under the condition that empty space is eliminated from Page 7 (top, middle), Page 8 (top, bottom), Page 9 (middle), Page 10 (middle), Page 11 (bottom), Page 13 (middle, bottom), Page 15 (middle) in the final version of the manuscript.
Author Response
Thank you!